

# Implementation of a Geological Disaster Monitoring and Early Warning System Based on Multi-source Spatial Data: A Case Study of Deqin County, Yunnan Province

CHEN Guo-ping[1, 2], ZHAO Jun-san[1*], YUAN Lei[3], KE Zun-jie[4], GU Miao[5], WANG Tao[5]

1. Faculty of Land Resource Engineering, Kunming University of Science and Technology, Kunming 650093, China;

2. Geomatics Engineering Faculty, Kunming Metallurgy College, Kunming 650033, China;

3. School of Information Science and Technology, Yunnan Normal University, Kunming 650500, China;

4. Yunnan Basic Surveying Technology Center, Kunming 650034, China ;

5. Kunming Yunjindi Geo-Information Co.Ltd., Kunming 650102,China)

*Correspondence author: ZHAO Jun-san (1429620189@qq.com)

**Abstract:** New technologies, such as three-dimensional laser scanning, interferometric synthetic aperture radar (InSAR), global navigation satellite systems (GNSSs), unmanned aerial vehicles (UAVs), and the Internet of Things, will provide greater volumes of data for surveying and monitoring as well as for the development of early warning systems (EWS). This research proposes solutions for the design and implementation of a geological hazard monitoring and early warning system (GHMEWS) for landslides and debris-flow hazards based on data multi-sourced from the aforementioned technologies. We describe the complex and changeable characteristics of the GHMEWS and analyze the architecture of the system, the composition of the multi-source database, the development mode and service logic, and the methods and key technologies of the system development. To illustrate the implementation process of the GHMEWS, we selected Deqin County as the case study area due to its unique terrain and diverse types of typical landslides and debris flows. First, we discuss the system's functional requirements and the monitoring and forecasting models of the system. Second, we examine the logic relations of the overall disaster process, including pre-disaster, disaster rescue, and post-disaster reconstruction, and develop a support tool for disaster prevention, disaster reduction, and geological disaster management. Third, we describe the methods for multi-source monitoring data integration and the generation and simulation of the mechanism model of geological disasters. Finally, we construct the GHMEWS for application to the dynamic and real-time management, monitoring, and forecasting of the entire hazard process in Deqin County.

**Keywords:** multi-source spatial data; geological hazard; monitoring and warning system; Deqin County





# 0. Introduction

China is a country that experiences a wide range of geological hazards with frequent and serious harm incurred due to the complex geological and geographical environment in most of its regions and the different temporal and spatial climatic conditions.[1] Geological hazards are the result of a dynamic process of environmental change[2] that can easily cause huge ecological disasters that threaten human life and property. Different types of geological hazards occur via different mechanisms. Even when the same types of hazard occur in different internal geological structures, the causes and characteristics of the environmental external terrain conditions of the hazard can differ. Therefore, currently, the catastrophe mechanism is yet to be grasped accurately, which makes it difficult to accurately predict the occurrence of hazards.

The rapid development of new generations of spatial information acquisition techniques, Internet of Things, video surveillance, and other technologies will provide a full range of multi-sourced data for the internal and external monitoring of geological hazards. Three-dimensional (3D) laser scanning, interferometric synthetic aperture radar (InSAR), high-resolution imaging, unmanned aerial vehicles (UAVs), global navigation satellite systems (GNSSs), and other technologies can quickly and accurately collect a large range of spatial location information. The Internet of Things and intelligent sensors can monitor the internal structure and microcosmic displacement changes of hazard bodies in real time and then rapidly transmit this change information. Video surveillance and rapid identification technology can track the exterior aspects of geological hazards and quickly identify the external changes and movement regularity, thus providing important information for early prediction and emergency response. Thus, it has become possible to conduct hazard mechanism analysis, generate hazard pre-warning and forecasting, simulate hazard processes, analyze hazard impacts, and conduct post-hazard assessment and planning to provide a strong data support and basis for decision-making by integrating massive data monitoring, constructing multi-source spatial databases, integrating catastrophic models, and establishing and developing the monitoring and early warning systems. As such, the development of a geological hazard monitoring and early warning system (GHMEWS) based on modern high-tech and multi-sourced data is receiving increasing attention from government departments and hazard prevention research institutions.

# 1. Review

The emergency characteristics and complexity of geological hazards make it difficult to integrate the monitoring and early warning methods and guarantee high reliability. In addition, there is no perfect theoretical system for the monitoring and early warning of geological hazards and there are many uncertain factors. As such, the technologies of geological hazard monitoring and early warning must be further researched.[3, 4]

The development of geological hazards monitoring and early warning is a long hard task. In recent years, researchers have used a variety of techniques for different types of geological hazard monitoring and early warning to





conduct relevant research and have achieved fruitful results. Bo[5] introduced the wireless sensor network in a
remote-monitoring system for coal mine safety based on Internet of Things. Hack[6] researched environmental
monitoring technologies and methods with respect to geological hazards in the Slovak Republic. Leung[7] instituted an
advanced monitoring system for debris-flow hazards. Kebaili[8] proposed an early warning system for landslide
hazards, which is based on the collection of big data by a wireless sensor network, including precipitation, soil
movement, and soil moisture data. This author also established and verified the MongoDB database prototype in the
Tunisian landslide high-risk area. Xie[9] and Liu[10] used GPS technology for monitoring the geological hazards. Yang[11]
introduced the principles, frameworks, and applications of 3S technology for geological hazard monitoring and
emergency command systems. Lei and Wang[12] and Lei et al.[13] researched the application of GIS technology in
geological hazard information management, hazard assessment, and system development from different angles and
obtained good results. In recent years, the use of the Internet of Things technology, wireless sensor networks, mobile
communication, 3D GIS, WebGIS, high-resolution remote sensing, UAVs, the BeiDou Navigation Satellite System,
and the multi-phase combination of geological hazard monitoring and early warning has gradually become mainstream.
However, the data obtained from each monitoring and early warning method has limitations and timeliness issues. The
development trends for future geological hazard monitoring and early warning forecast systems include the integrated
use of new mapping technologies, video surveillance, and the Internet of Things technology to quickly access
multi-source data, accurately access internal micro-displacement and external geological hazard catastrophes from a
wide range of macro-spatial location. This is achieved by combining ultra-long-range microwave signal real-time
transmission technology with cloud computing and big data mining algorithms for the real-time monitoring of
geological hazards and building integrated information systems for dynamic simulation, early warning, and
decision-making.

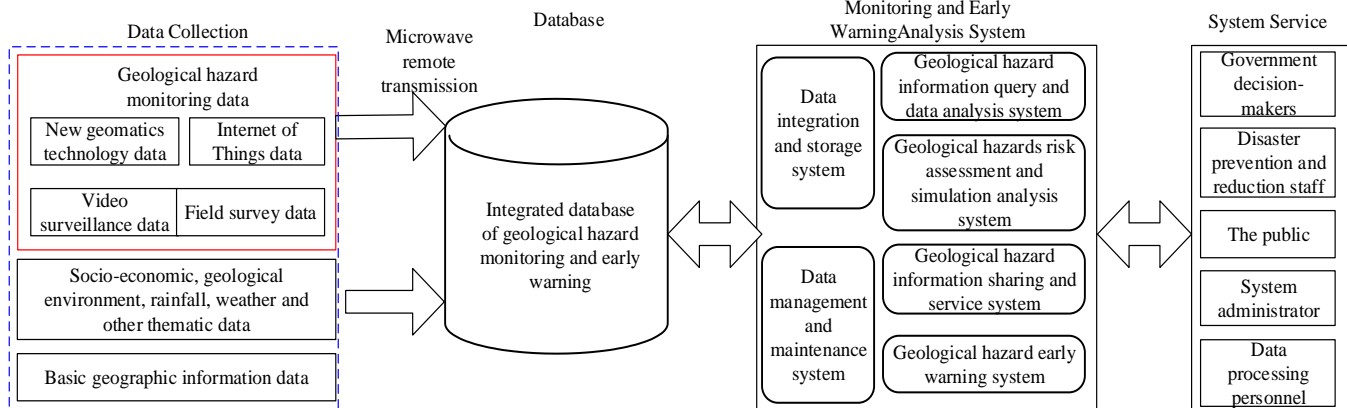

**Fig. 1 Structural map of the geological hazard monitoring and early warning system (GHMEWS)**



## 2. Design of the GHMEWS Based on Multi-source Data

### 2.1 Design of System Structure

The GHMEWS based on multi-sourced spatial data comprises four parts (Fig. 1). The data acquisition layer is based on existing basic geographic information data of the study area as well as geological environment and meteorology data, which combines data from the new geomatics technology, the Internet of Things technology, video surveillance, and real-time monitoring information. The database layer classifies and layers the collected geological hazard-monitoring information according to the spatial database constitution standard for building the GHMEWS core database.[14] The monitoring and early warning analysis system platform layer is based on multi-source monitoring data, combined with the professional model, thus providing subsystems for actual demand. The system service layer shows different platforms to different users based on the processing results of the analysis system.

### 2.2 Database Design

In accordance with uniform standards and norms, the database collects and integrates multi-type and multi-scale basic geographic information data in a unified manner and organizes and effectively stores data generated by multi-period 3D laser scanning, UAV, video surveillance, intelligent sensors, and geological hazard field surveys to ensure the horizontal integrity of the survey data and the multi-sourced data to be superimposed and spatially combined by transforming coordinates. According to the database requirements and standards, the data is integrated and the integrated management of the spatial data includes survey graphics, attributes, images of the study area, and other non-spatial data.

Database construction mainly includes the following aspects:

(1) Geographic Conditions and Basic Geographic Information Database

Based on the general-survey database of the geographical conditions of the study area, existing multi-source, multi-scale, multi-temporal, and four-dimensional (4D) products and related geographic information data are integrated to define the geographic conditions and assemble the basic geographic information data of the study area as a systematic original base map.

(2) Special Socio-economic Environment Database

Data regarding the geological structure and environment, long-term geological hazards and rainfall, geological hazard control projects, large-scale economic and social maps, buildings, and unit personnel distributions in the study area are collected to establish this socio-economic environment thematic database.

(3) Geological Hazard-Monitoring Database

The monitoring data for the same hazard locations or regions include 3D laser-scanning data, UAV data, video surveillance data, Internet of Things data, InSAR data, and geological field survey data. Thus geological hazard-monitoring database is established in chronological order.

### 2.3 Design of Development Model



The GHMEWS is based on a data normalization and standardization process and relies on a computer network and hardware platform. Its core stores scattered data and centralizes usage. This model was developed using GIS technology, big data mining, and information security.[15] It can integrate geological hazard-monitoring data, queries, statistically analyze data, and multi-dimensionally and rapidly display hazard information, perform risk assessment and modeling, share information, and perform early warning and forecasting.


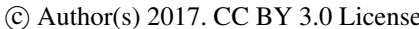



**Fig. 2 Diagram of the system development model**

This system adopts a C/S and B/S mixed development mode. Based on the C/S model, this system mainly provides data integration and storage, geological hazard risk assessment and simulation analysis, and geological hazard

early warning and forecasting. The B/S mode is mainly used for geological hazard information queries and data analysis, geological hazard monitoring and information-sharing services, and data management and maintenance to meet the various application needs. The services are provided to all departmental staffing levels of land and resources management, hazard prevention and mitigation, and governmental decision-making. The process is implemented as follows:

(1) Data flow: Geological hazard-monitoring data is uploaded to the data center server through the data integration and storage interface. Then, the data-processing staff converts the data format, edits the graphics, and performs other processing tasks. The system administrators then review the data and release it to the frontend of the application system.[14]



(2) Technical realization: The system adopts the C/S model based on the GIS service and performs functions such
as data processing, management, transmission, addition, deletion, checking, and changing. The backend system comprises several subsystems with distributed five-tier structures according to need. The frontend system provides daily management services and performs functions related to information browsing, queries, and spatial analysis. It adopts the relatively mature .NET framework, and users can access it using their browser.

(3) Network structure: The core database involves confidential data and runs in a classified network. The related
data management software also runs in the classified network and directly accesses the database. The application server is responsible for visiting the core database and providing specific business. At the client end, the application server can be directly accessed. This structure is based on two important factors. The first factor is associated with the security and confidentiality requirements. It separates the internal and external networks and limits the connection setting. The second factor is based on the system requirements. . The real-time maintenance of the core database is
performed via the data management and maintenance system.

## 2.4  Design of Business Logic

The system is divided into five levels. The bottom level is the big database system used to store a variety of spatial and non-spatial data. It provides data support for the business logic run program through data engine middleware. In addition, it provides operations and applications for all types of users through the application
subsystem. Software development relies on a variety of development tools, middleware, and GIS secondary development tools.[14] Figure 3 shows the design of the business logic structure of the system.

The logical structure of the system is divided into several parts, including the database system, data engine middleware, business logic development middleware, application system, and user operation terminal.

The database system, which is responsible for direct data interaction, provides the specific data operation of the
database. The data engine middleware mainly provides an intermediate medium whose program interacts with the database through the data engine. To enhance the security of the database, the data engine is used to reduce coupling of the program and database operations. The business logic development middleware provides some basic component modules to increase the relevance of application systems in these modules and technologies. The operating system works mainly through the various components and related technologies in an organic combination according to the
specific functions of the system. User operations provide pertinent business services for each user in accordance with the structural setup, each layer of work, convenience of maintenance, functional time expansion, and user-friendly features.




**Fig. 3 Design of the system's business logic structure**

# 3. Technology and Key Technologies

### 3.1 System Functions

The main task of the GHMEWS based on spatial multi-source data is to monitor the temporal and spatial evolution of geologically hazardous areas as well as the internal and external changes in hazard areas in real time by various technical means to obtain the maximum amount of continuously changing data associated with the hazard and corresponding environmental factors. In addition, this integrated professional model establishes a dynamic monitoring




model with access to dynamic ground-hazard assessment results for application to the early warning and prediction of hazards. This system performs several functions: data integration and storage, information query and data analysis, data management and maintenance, information sharing and services, risk assessment and simulation analysis, and early warning (Fig. 4). For hazard prevention and mitigation departments, the public, and policy makers, it offers the ability to develop the pre-hazard prevention, hazard relief, and post-hazard planning and management strategies by providing critical data support and a strong basis for decision-making.

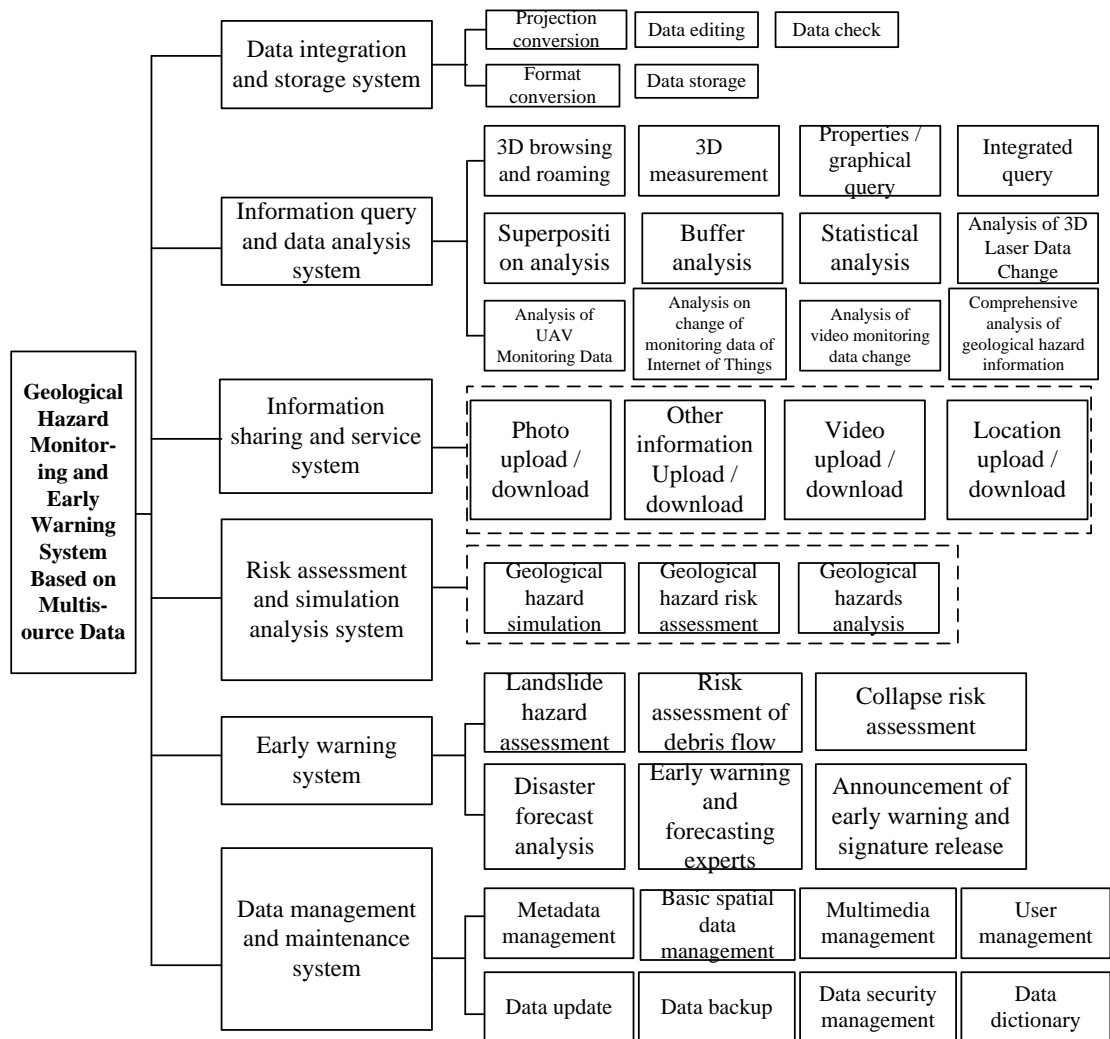

**Fig. 4 A chart of the system functions**





## 3.2 Key Technology

### 3.2.1    Multi-source Monitoring Data Integration and Integration

This platform relies on a multi-level, integrated geospatial and attribute database, whose data sources are extensive and have multi-source, multi-GIS, and database platform features. To develop this information system, it is necessary to integrate the scattered and heterogeneous data with different formats into an "information silo" to integrate and standardize the management of these information resources.[16] In addition, with the development of commercial database technology, the integrated data storage and management of spatial and associated attributes has become possible;[17] thus, the integrated management of spatial and attribute data and different types of basic database data integration management models is an inevitable trend. Therefore, the multi-source monitoring data is mainly integrated to lay a foundation for later data exchanges, sharing, and co-processing between departments, industries, and platforms.

### 3.2.2    Fast Automatic Recognition and Early Warning of Geological Hazard Information Based on Video Surveillance

Based on the video monitoring of geological hazards, the key automatic identification and early warning technologies have the following two aspects.

First is the automatic and rapid identification of hazard status changes and alarm thresholds. We use a high-resolution video probe to continuously detect geological hazards, and the continuous video images are automatically aligned at various times to identify the micro-deformation process and development trend of the hazard. When the frontend hazard area deformation exceeds the established warning threshold, an alarm signal is sent to the total console and users can display in a timely manner the alarm area, location, status, monitoring equipment, and warning level in a timely manner via direct text messages and images to persons in charge and the executive leadership.

Second, this technology uses microwave transmission for larger volumes of video-monitoring data as the traditional communication network is slow. In addition, geological hazards are often located in the mountains, where there are frequent network signal blind spots. Microwave wireless networks can achieve the fast big data transmission of specific monitoring areas and voice communications. In addition, the microwave wireless network installation is convenient, flexible, scalable, and has considerably low operating costs compared with expensive GSM and other communication systems. By using the monitoring equipment to collect data and GIS data to match, we can obtain information coordinates to quickly monitor the heat source and accurately determine the monitoring location.

### 3.2.3    Mechanism Model of Geological Hazards and Simulation and Analysis Technology

Due to the geological hazards associated with mass movement and induced by other phenomena,[2] the core factor for consideration is the geological structure and defects associated with hazard incentives, including natural activities, such as earthquakes and rainfall, and artificial activities, such as mining, excavation of the slope foot, chaotic cutting,



and other irresponsible human activities. The diversity of the influencing factors makes it difficult to accurately

determine the hazard mechanism. It is critical to conduct long-term monitoring of the characteristics of the hazard body deformation and pre-disposing factors by using 3D/4D GIS and VR technologies, such as a large number of multi-source monitoring data analyses, extracting the catastrophe principles, determining the hazard mechanisms and key inducements, and constructing an overall mechanism model of geological hazards. On this basis, we must integrate regional rainfall, weather, geological environment, and real-time data, quantitatively analyze and simulate the

catastrophic trends, and provide services for accurate forecasting of geological hazards and project management design.

### 3.2.4 Construction of Internal and External Integrated Monitoring System With the Internet of Things and New Geomatics Technologies

Internet of Things is a network based on the Internet, the traditional telecommunications network, and other

information carriers to achieve the common physical interoperability of objects. All objects are connected through the Internet as is the information-sensing equipment for the exchange of information, i.e., the objects of interest, to achieve intelligent identification and management.[18] With this technology, we can set a large number of physical targets and deformation sensors in the hazard target area and create a sensor network to collect and transmit micro-displacement information about the hazard body in real time. The external deformation monitoring of the regional hazard body relies

on modern mapping technologies, such as 3D laser scanning, GNSS, and UAVs. The monitoring data obtained by these two technologies differ in their data structure, format, and dimensions. Therefore, key issues are how to enable time synchronization and develop spatial references for a unified internal and external integrated monitoring system for the integrated application of multi-dimensional heterogeneous monitoring data.

## 4. Experiment and Conclusions

### 4.1 Experiment Analysis

Based on the system platform construction concept, we selected Deqin County as the study area. Deqin County is located in the Yunnan Diqing Tibetan Autonomous Prefecture, which is a mountainous area, and the county and the surrounding area is a complex geological environment. Currently, the county planning area includes the Zhixi River, Shuimofang River, Yizhong River, and Jushuihoushan River gully, four debris flows with 95 landslide positions, five

unstable slopes, and eight collapsed (or peeling) areas in the Yunnan Province, which represents one of the country's most serious geological hazards in this relatively hidden county. However, due to various considerations, the county cannot be relocated. Hence, a variety of technical means must be implemented to monitor and control its geological hazards and provide real-time early warning for minimizing the impact of the geological hazards. As such, the establishment of an overall Deqin County GHMEWS has become an urgent task. The development and construction of

the GHMEWS will play a crucial role in the comprehensive prevention, reduction, and control of geological hazards




and in its regional economic development, social stability, and improvement of people's living standards. In addition, the types of geological hazards in this region are diverse and the topography and geomorphology are unique, with typical representative demonstration significance.

The GHMEWS of Deqin County relies on existing geographic survey data, geological hazards, and special monitoring data, and integrates socio-economic environment, geological environment, and historical data for constructing its core database of geological hazard monitoring and warning. Based on geological dynamics and catastrophe theories, we can establish a model of geological hazard forecasting on which the GHMEWS is based, which includes the multi-source data integration and storage, information query and data analysis, data management and maintenance, information sharing and service, risk assessment, and simulation analysis and early warning 255 functions. It also provides technical support in the form of pre-hazard geological data, hazard rescue information, post-hazard reconstruction information, and hazard management strategies. Figures 5–7 show the main interface of the management system, the interface of the geographic information system, and the function of the protection and care subsystem, respectively.

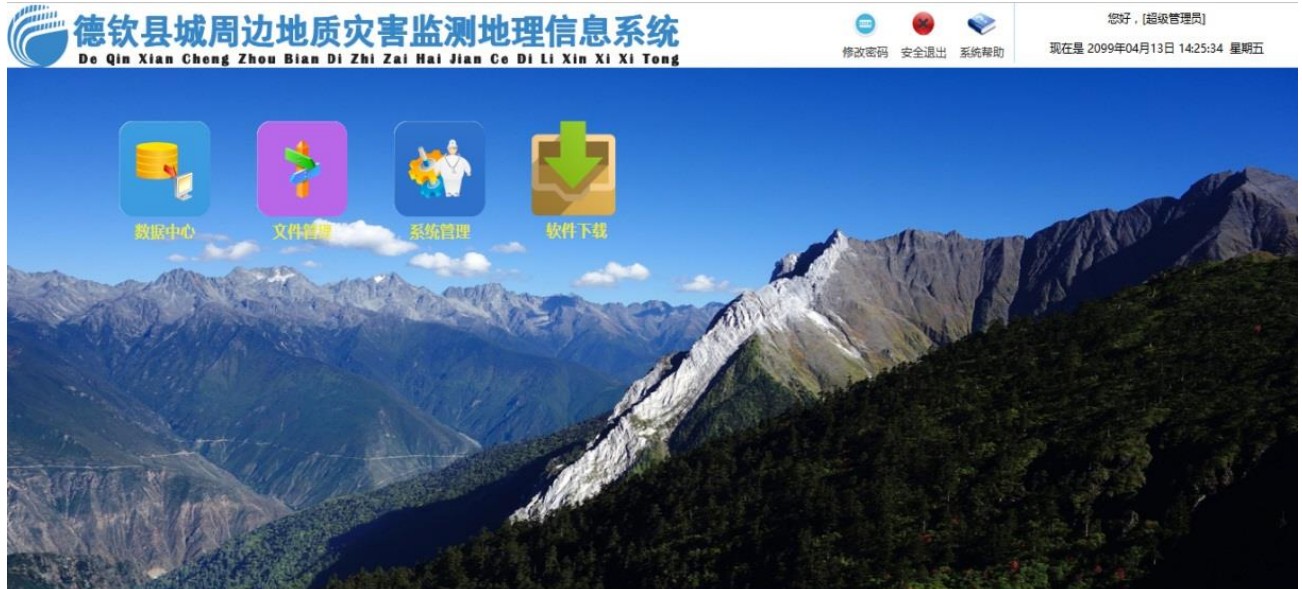

**Fig. 5 Management interface of geological hazard monitoring and early warning geographic information**
**system**





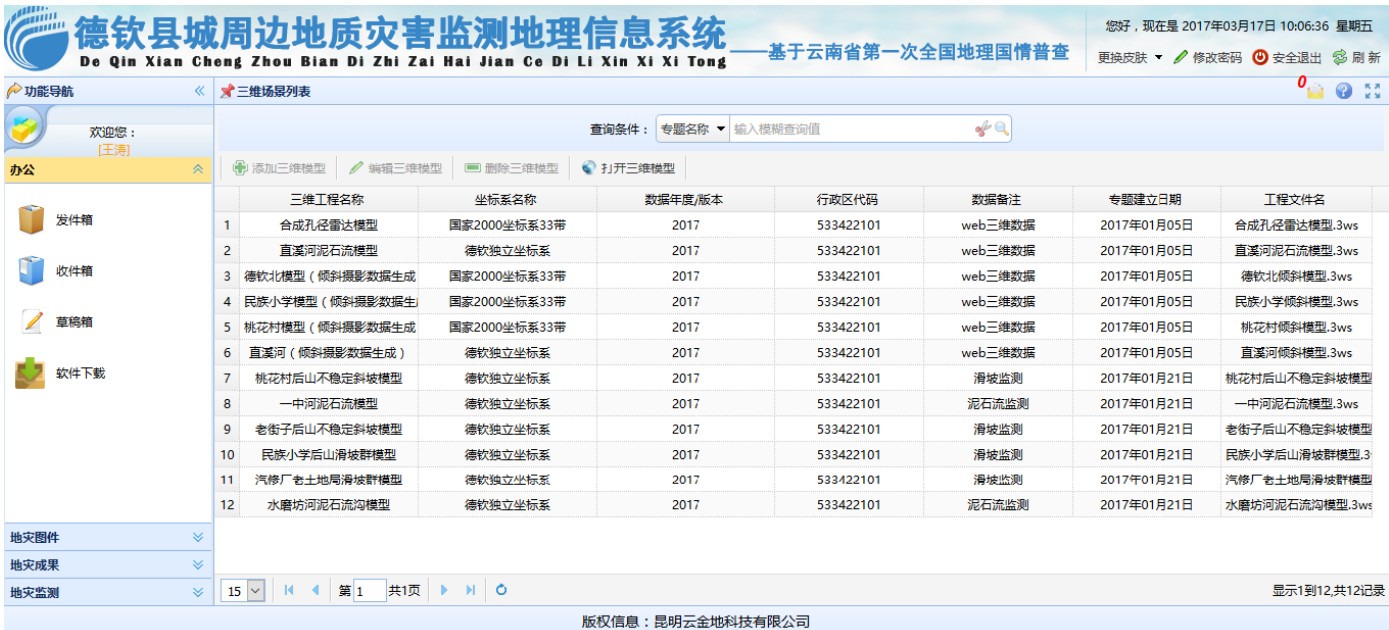

**Fig. 6 Main interface of geological hazard monitoring and early warning geographic information system**

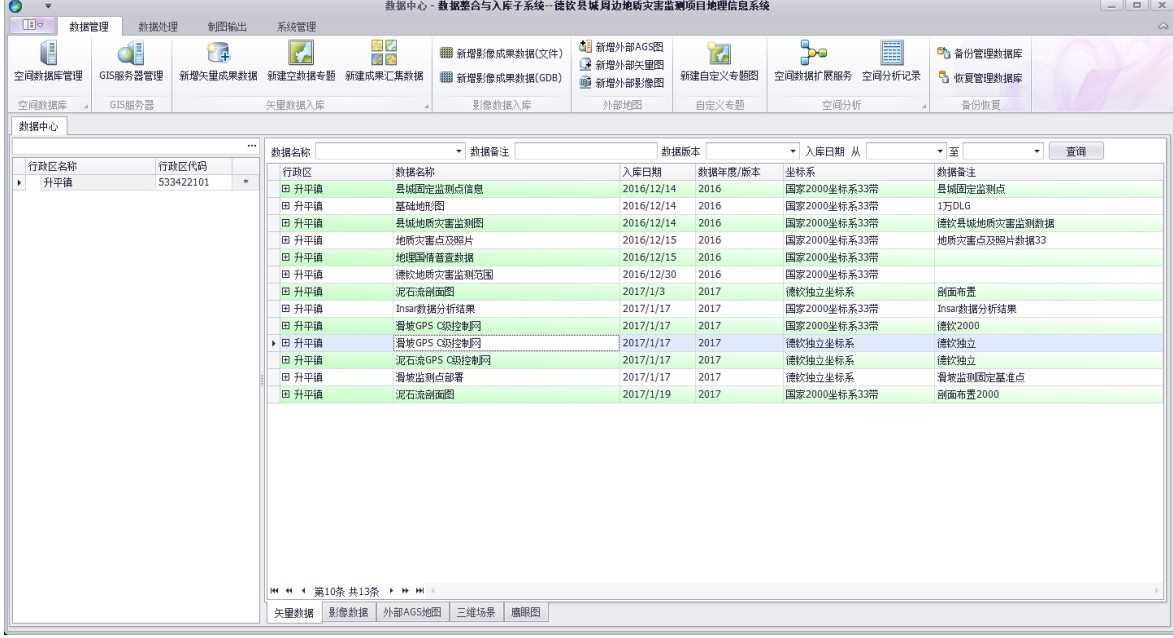

265                       **Fig. 7 GHMEWS data management and maintenance subsystem**



The pre-hazard warning and forecasting system, based primarily on the rainfall and meteorology data for the hazard area from data mining and spatial analysis, predicts hazards beforehand and estimates the pre-hazard status and hazard levels so that hazard prevention and mitigation measures can be planned in advance to minimize losses.

In the geological hazard process, based on real-time monitoring data, we can quickly assess the risk of geological hazards, predict the evolution, simulate the hazard development trend, and analyze the influence range of the hazard points to provide a decision-making basis for the road system, route analysis, rapid material arrival, population evacuation, and other rescue data.

After the occurrence of a hazard, the system provides summary statistics and an analysis of the affected areas, 275 populations, losses, and other conditions in real time to support decision makers in post-hazard reconstruction planning.

Based on the big data of geological hazards, we can analyze the mechanism and influencing factors of different types of geological hazards and identify the key factors of different types of geological hazards to implement various control and engineering management measures.

In brief, the system platform provides data support related to a variety of geological hazards for hazard prevention and mitigation. The actual demand for hazard prevention and mitigation strategies drives the specific R&D content of this system, and as such, they are inseparable.

## 4.2 Conclusions

Considering the shortcomings of traditional hazard monitoring and early warning methods, due to limited data 285 sources, lack of data integrity, and lack of real-time capability, in this research, we analyzed the modern spatial data acquisition and Internet of things technologies to propose a new-generation GHMEWS based on multi-sourced spatial data and a technical framework and development model of an early warning and forecast system. We examined system technologies and proposed key technical problems that are required to be resolved. As a reference, we used Deqin County to consider the functions of information query and analysis, risk assessment and simulation, geological hazard 290 early warning and forecasting in the GHMWES based on multi-sourced spatial data. To a certain extent, this alpine valley of geological hazards is a good example showing the need to install monitoring and early warning intelligence to enhance the degree of dynamic information available for similar high-risk areas. The construction of the GHMEWS involves complex systems engineering and diverse technologies. There are many areas that are yet to be investigated with respect to hazard data processing and analysis, big data integration and mining, hazard process simulation and 295 prediction, and real-time transmission using microwave signals. We plan to continue our research and development in this field to achieve more mature and effective results.





## 5. ACKNOWLEDGEMENTS

We express heartfelt thanks Yunnan Provincial Science and Technology Department and Kunming Science and Technology Bureau provide financial support for this research.

Part funding for the software implementation for this study was provided by the Yunnan basic surveying and Mapping Technology Center. Part funding for the data processing and model development was provided by Kunming YJD Co., Ltd.

We are greatly thanking EGU conference for give the opportunity to us attend the meeting, discuss and communicate issue with some expert and participant. At the same time, we are greatly indebted to reviewer for his valuable instructions and suggestions on our thesis as well as his careful reading of the manuscript.

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
