# Peer review of "Implementation of a Geological Disaster Monitoring and Early Warning System Based on Multi-source Spatial Data: A Case Study of Degin County, Yunnan Province"

_Natural Hazards and Earth System Sciences, 2017_

## Referee Comment (RC1) · Anonymous Referee #1 · 23 Aug 2017

Dear authors;

I have been invited to review your manuscript titled "Implementation of a Geological Disaster Monitoring and Early Warning System Based on Multi-source Spatial Data: A Case Study of Deqin County, Yunnan Province" submitted to NHESS. Manuscript deals about an interesting subject; however based on my review, I suggest some revisions. Please find my specific comments and recommendations in the following:

Section 0: The china geological hazards are explained, but there are not any details

and statistics about them on the context. In the other words, the importance of the topic is not well described. Additionally, you did not provide enough references in this part. Also, the innovation aspect of GHMEWS needs to be explained in comparison to similar systems in the world.

Section 2: The procedures for obtaining data should be explained more.

Page 5: In this article, C/S and B/S models are used but the explanation about them are not enough and it is necessary to mention several references.

Section 3:The methodology of combination and weighting information, assessment the risk of geological hazards, simulation the hazard development trend, etc. is very poorly explained.

Section 3:The key factor for identification alarm thresholds, the radius of the blind zone (where no warning can be given because the processing time is too great to send a warning before the geological hazard starts), and the amount of elapsed time for alerting warning must be explain.

Section 4: The results of GHMEWS implementation for mentioned case study has been not clearly shown.

Fig.2: The figure is unclear and difficult to read. Please improve the legibility of it as far as possible.

Finally, sometimes the reading of the paper quite hard. So I advise the author to correct and improve it preferably by a native speaker.

---

## Referee Comment (RC2) · Anonymous Referee #2 · 12 Oct 2017

The development of Landslide Early Warning Systems and Spatial Decision Support Systems for landslide risk management is an interesting topic, and this paper aims to present work on the development of such a system in China. Nevertheless the paper remains rather vague and superficial, and never really gets in depth about the functions of the system. This may be partially contributed by the often confusing manner of writing. The aim is to present how different sources of information (e,g, LIDAR, INSAR, UAV, GPS, smart sensor networks etc) can be used for landslide monitoring, early warning, hazard and risk assessment etc. However, the paper does never present how

these data are to be used. For instance how landslide inventories are generated using these tools, what type of landslide hazard assessment is carried out , whether landslide runout analysis is done, how landslide intensity is expressed, how vulnerability is quantified, how uncertainty is incorporated in the risk assessment. The paper only describes the structure of the system is different ways, which often contain repetitions. For example, the first three figures seem to be indicating the same , but in different form. When the example is given from a specific study area (Deqin county) the paper remains vague, and so do the figures. Figures 5, 6 and 7 do not contain any useful information for non Chinese speakers, and I have asked it to a Chinese colleague, also not for Chinese speakers. The authors should properly explain how the data was used for monitoring landslides (were these slow moving landslides, or accelerating ones), how debris flows were predicted, and how successfully this was done. The introduction is confusing and many of the sentence are difficult to follow or seem to be contradictory. For example "temporal and spatial climatic conditions", "geological hazards are the result of a dynamic process of environmental change that easily cause huge ecological disasters", "the catastrophe mechanism is yet to be grasped accurately", "internal and external monitoring of geological hazards", "a large range of spatial location information", "microcosmic displacements", "post hazard assessment", "catastrophic models" etc. The literature review is not very exhaustive and seems to be anecdotal, and several of the papers are already quite old, whereas the technology is rapidly developing. The authors should explain better why it is still very difficult to make accurate landslide early warnings, as well as hazard and risk assessments. The authors don't seems to make a difference between local LEWS that measure actual displacements, and more regional precipitation based systems. I don't agree with the statement that "multi-phase combination of geological hazard monitoring and early warning has gradually become mainstream". Section 2 on the the design of the GHMEWS is often difficult to follow and seems to be full of isolated statements. Why do you need to combine geographic information with four dimensional products? Why are geological data part of the socio-economic database? What is the design of a development model? What are B/S and

C/S mixed development modes? And why does geological hazard risk assessment and simulation analysis belong to C/S and geological hazard monitoring to the other? What is the design of a business logic? What is a business logic structure of the system? What is middleware? Why do you combine all kind of generic tools like drawing and printing with specific ones such as risk assessment in Figure 3? Section 3: what is the difference between technology and key technologies? Also in this section you repeat the general things, but do not get into more depth. The experiment and conclusions in section 4 are too general and do not go into depth.

---

## Author Comment (AC1) · 2 Jan 2018

Thank you very much for your comments about our paper submitted to nhess-2017-191. We have learned much from those comments, which are fair, encouraging and constructive. After carefully studying the comments and your advice, we will make corresponding changes. We are so please to receive those comments about the paper.